# Evidence-driven spatiotemporal COVID-19 hospitalization prediction with Ising dynamics

Junyi Gao [1,2], Joerg Heintz [3], Christina Mack[4], Lucas Glass[4], Adam Cross [5] ✉ & Jimeng Sun [3] ✉

In this work, we aim to accurately predict the number of hospitalizations during the COVID-19 pandemic by developing a spatiotemporal prediction model. We propose HOIST, an Ising dynamics-based deep learning model for spatiotemporal COVID-19 hospitalization prediction. By drawing the analogy between locations and lattice sites in statistical mechanics, we use the Ising dynamics to guide the model to extract and utilize spatial relationships across locations and model the complex influence of granular information from real-world clinical evidence. By leveraging rich linked databases, including insurance claims, census information, and hospital resource usage data across the U.S., we evaluate the HOIST model on the large-scale spatiotemporal COVID-19 hospitalization prediction task for 2299 counties in the U.S. In the 4-week hospitalization prediction task, HOIST achieves 368.7 mean absolute error, 0.6 $R^2$ and 0.89 concordance correlation coefficient score on average. Our detailed number needed to treat (NNT) and cost analysis suggest that future COVID-19 vaccination efforts may be most impactful in rural areas. This model may serve as a resource for future county and state-level vaccination efforts.

The COVID-19 pandemic has caused enormous social and economic loss. With over 90 million confirmed cases and 1 million deaths in the U.S. by Aug 2022[1], the pandemic placed a heavy burden on national and global healthcare systems. The disparities in medical resource availability among U.S. counties—including ventilators, hospital beds, and critical care staff—can and, in some cases, have devastatingly impacted patient outcomes[2]. Nationwide vaccination efforts have primarily favored urban areas, and the urban-rural disparity in vaccination coverage has continued to worsen throughout the pandemic[3]. Rural vaccination efforts are technically challenging due to a combination of more limited access to healthcare, generally lower perception of the severity of COVID-19 by residents of rural communities, and usually higher levels of vaccine hesitancy[3]. This multi-year trend of relative rural under-immunization raises the question of whether further vaccination efforts might be more impactful in these under-

immunized areas regarding hospitalization, death, and cost-effectiveness. Answering this question is technically challenging but clinically meaningful; if counties can be stratified in terms of improved outcomes per vaccination, this may provide meaningful insight into how local and statewide efforts might be distributed to maximize impact. To accomplish this task, we propose the following technical contributions:

1. Extracting inter-region similarities. The pandemic progression might be similar in regions with similar geographical and socioeconomic properties. If so, extracting and utilizing these similarities would help the model better predict pandemic progression in the future. Most existing spatiotemporal pandemic prediction methods use pre-defined location graphs to explicitly inform the spatial connections to the model[4–7]. In contrast, these manually defined location graphs may differ significantly from the

[1]The University of Edinburgh, Edinburgh, Edinburgh, UK. [2]Health Data Research UK, London, UK. [3]University of Illinois Urbana Champaign, Champaign, IL, USA. [4]IQVIA, Durham, North Carolina, USA. [5]University of Illinois, College of Medicine Peoria, Department of Research Services, Peoria, IL, USA. ✉ e-mail: arcross@uic.edu; jimeng@illinois.edu

ground-truth inter-regional relationships. Other works learn the spatial connectivity purely based on the historical pandemic progressions[8,9], regardless of the underlying social, economic, or geographical similarities, which cannot capture the inter-region similarities in full view and lead to inferior prediction results. Therefore, it is non-trivial to model spatial dependencies with multi-source background data in a flexible way.

2. Utilizing complex inner-region influence factors. Local COVID-19 hospitalization rates are likely affected by many factors, including the number of infected patients, high-risk patient population size, and vaccination rates. Estimating these effects becomes more challenging with more granular information, such as the number of COVID-19 immunizations per individual. Leveraging these complex inner-region influence factors, such as medical claims and vaccination statistics, may significantly improve hospitalization prediction performance. Recent research combines the susceptible-infected-removed (SIR) and susceptible-exposed-infected-removed (SEIR) differential models with deep learning models to improve predictions by simulating real-world SIR dynamics[4–7]. However, when applied to hospitalization prediction tasks, these models face the following issues: (1) Limited data sources and model parameters in the traditional epidemiological model make it difficult to utilize these complex high-dimensional inner-region factors effectively. Ignoring complex inner-region factors may cause the model to fail to capture the underlying disease distribution, virus subvariants, and vaccination effectiveness rates across the state, which could lead to biased results. While purely deep learning-based time-series prediction models such as recurrent neural networks (RNN) can handle high-dimensional data and extract complex nonlinear relationships, they ignore real-world progression dynamics and have minimal interpretability compared to epidemiology-based models. (2) Most epidemiological models offer a simple increasing-decreasing trend with different rates, while the hospitalization curves are more complex and may not follow this simple trend.

3. Moreover, the SIR dynamics are not designed for spatiotemporal prediction tasks, so these models cannot directly utilize for hospitalization prediction.

To address the above challenges, we propose an Ising dynamics-based deep learning model for spatiotemporal COVID-19 hospitalization prediction. We have combined multiple data sources, including disease and vaccination statistics from real-world medical claims, medical resource usage, census, and geographical and mobility data, to create an evidence hub for the ability to train a complex evidence-driven spatiotemporal prediction model, HOIST. The proposed HOIST model can handle complex inner-regional influence factors from multiple data sources and adaptively learn the inter-regional relationships using the locational census and geographical and inter-locational mobility data without requiring a pre-defined location graph with fixed edges.

The model learning process is guided by the Ising dynamics, a mathematical statistical mechanics model to estimate site spin configurations in square lattices[10]. By drawing the analogy between locations and lattice sites, we find the Ising dynamics a natural choice to model the inner-region factors and inter-region similarities simultaneously: (1) the Ising dynamics guide the model to extract and utilize inter-region spatial relationships by taking prediction results from similar locations as kinetic energy; (2) the Ising dynamics model the complex influence of granular inner-region factors from the real-world clinical evidence as potential energy (i.e., external fields). Both kinetic energy and potential energy jointly decided the energy of the location (predicted hospitalization case to increase or decrease). The Ising dynamics in this paper are used as a regularizer to guide the model learning and prediction process, which enables the model to learn complex nonlinear patterns flexibly. We use recurrent neural networks (RNNs) to learn the parameters of the Ising model, so our model can also extract and utilize temporal patterns in the data.

The overview of HOIST is shown in Fig. 1. The HOIST model first uses static data to calculate the distances between locations in the latent space. These distances are further normalized into 0 and 1, which indicate the connectivity between locations learned by the model. The dynamic data are used to estimate the external fields (EFs) of locations and used to generate predictions with LSTM. The estimated EFs and the learned connectivity are used to calculate the Ising dynamic loss, which is then used to regularize the prediction results and better model the real-world connectivity and effects of various influence factors.

We evaluate the HOIST model on the large-scale spatiotemporal COVID-19 hospitalization prediction task for 2299 counties in the United States. This scale is much larger and more granular than existing COVID-19 spatiotemporal predictive works. In the 4-week hospitalization prediction task, HOIST achieves 368.7 MAE, 0.6 $R^2$ and 0.89 CCC scores. We also conduct experiments with different lengths of prediction windows from 1 to 5 weeks. Compared to the best baseline model, HOIST achieves 48% lower MAE, 65% lower MSE, 272% higher $R^2$ and 51% higher CCC on average. The prediction performances under the temporal data split setting show that HOIST can consistently achieve low prediction error regardless of the underlying data distribution shift caused by new virus variants. These results suggest the HOIST model can accurately predict both long-term trends and short-term variations, enabling broader real-world applications.

Unlike totally black-box deep learning time-series models, the fusion of real-world dynamics in HOIST allows us to look deeper into the model to see how various influence factors affect the model predictions. By analyzing the weights of external fields, we find that the booster vaccination rate has a more significant negative correlation with future hospitalization cases (i.e., more profound impact) compared to the first and second vaccinations in the series. We also find that the effects of two major vaccination brands, Pfizer and Moderna, have no statistical difference in our model. By simulating varying immunization rates, we conduct a detailed analysis of the marginal benefit of the vaccination ratio by answering two questions: (1) How many more vaccinations are necessary to prevent one hospitalization case for a specific location? (2) What is the cost ratio between these vaccinations and the average COVID-19 hospitalization for a specific location? Quantitatively, our model shows that increasing the vaccination ratio by 10% can reduce the number of current hospitalization cases by 15% on average for all locations. We also find that the cost ratio is generally highest in much of the rural Midwest and Rock Mountain regions, suggesting that prioritization of vaccine efforts in these counties could most significantly reduce the overall statewide healthcare financial burden of COVID-19. For 368 counties among 43 states all over the United States, rural vaccination outreach efforts are likely cost-saving endeavors. We believe these results and our model may inform clinicians, healthcare institutions, and policymakers to improve their decisions and ultimately reduce the negative economic and health impacts caused by the pandemic. All detailed county-level analysis results are available at the online visualization platform at https://v1xerunt.github.io/HOIST/.

## Results
### Problem formulation and data sources
We develop the HOIST model to predict the total number of COVID-19 hospitalization cases in an upcoming 4-week period at a county level across the United States. Throughout this paper, we use $N$ to denote the number of locations (counties) and $T$ to denote the length of timesteps (days). The model uses static location background data to adaptively learn location connectivity and then uses dynamic data to learn the temporal patterns. The model uses the learned

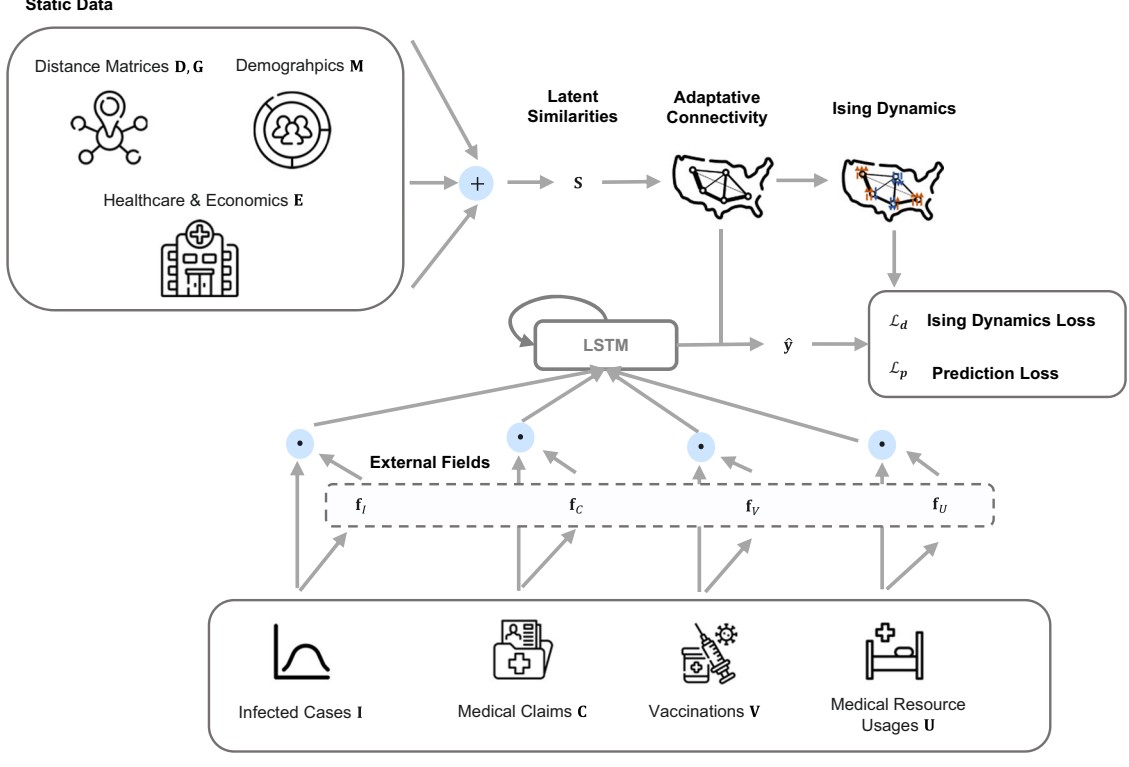

**Fig. 1 | The proposed HOIST model.** We use static data to calculate the latent distances between locations. The latent distances are then used to calculate the adaptive connectivity of the location graph. The dynamic data are used to estimate the external fields (EFs) and then generate predictions using the long short-term memory network (LSTM). We use the Ising dynamics to regularize the spatiotemporal prediction results using the estimated external fields (EFs) and the adaptive connectivity.

spatiotemporal patterns to conduct predictions. Concretely, we formulate model inputs and the prediction task as follows:

**Input 1 (static data).** The static data include distance matrices $\mathbf{D}, \mathbf{G} \in \mathbb{R}^{N \times N}$, population demographics statistics $\mathbf{M} \in \mathbb{R}^{N \times 9}$ and economics and healthcare statistics $\mathbf{E} \in \mathbb{R}^{N \times 4}$. $\mathbf{D}$ is the geographical distance matrix and $\mathbf{G}$ is the mobility distance matrix. The geographical distances are the Haversine distances between $N$ locations. The mobility distance includes the average mobility flows between $N$ locations during 2020 and 2021. The mobility scores are collected from the Multiscale Dynamic Human Mobility Flow Dataset, which analyzed millions of anonymous mobile phone users' visits to various places provided by SafeGraph[11]. The census features $\mathbf{M}, \mathbf{E}$ include populations of different age and race groups and medical resource statistics, which are collected from the county-level census dataset provided by ref. 12.

**Input 2 (dynamic data).** All dynamic data are three-dimensional tensors, including the daily new infected COVID-19 case counts $\mathbf{I} \in \mathbb{R}^{N \times T \times 1}$, medical claim statistics $\mathbf{C} \in \mathbb{R}^{N \times T \times 20}$, vaccination statistics $\mathbf{V} \in \mathbb{R}^{N \times T \times 17}$, and medical resource usage statistics $\mathbf{U} \in \mathbb{R}^{N \times T \times 4}$. The daily new cases $\mathbf{I}$ are collected from the Johns Hopkins COVID-19 Data Repository[13]. Claim statistics $\mathbf{C}$ include daily statistics of total patients, patients older than 65, and patients with certain comorbidities. The age threshold and comorbidities are derived from the CDC COVID-19 guidelines[14] and the Charlson Comorbidity Index[15]. The vaccination statistics $\mathbf{V}$ include daily statistics of 13 different vaccination CPT codes. Both vaccination and claim statistics are collected from IQVIA's real-world claims dataset[16]. The medical resource usage statistics $\mathbf{U}$ include the usage of inpatient and ICU beds and their usage for COVID-19 patients, which are collected

from HealthData.gov[17]. Though $\mathbf{U}$ also provides information for hospitalization cases, we use $\mathbf{C}$ instead of $\mathbf{U}$ as the ground-truth case numbers. This is because $\mathbf{U}$ only provides weekly statistics, and the statistics are collected from fewer healthcare institutions compared to the claims data. We include medical resource usage statistics $\mathbf{U}$ to help reduce the potential biases in the claims data.

The feature list and statistics can be found in Supplementary Tables 1–4.

**Task (spatiotemporal COVID-19 hospitalization prediction).** Given all static and dynamic data of $N$ locations and $T$ timesteps, our task is to predict the total number of COVID-19 hospitalization cases in future $L$ days for each of the $N$ locations, denoted as $\mathbf{y} \in \mathbb{R}^{N}$. Since the claims data do not show the primary cause for hospitalizations, we define a hospitalization case as a patient hospitalized within 35 days after the COVID-19 diagnosis in the claims dataset, and we calculate the total hospitalization cases for each location to get $\mathbf{y}$. This setting is inspired by previous COVID-19 hospitalization prediction works[18,19].

### Experiment settings
In the experimental phase, we extract all the required data from September 2020 to May 2022. We set the input window $\tau$ to 5 weeks and the prediction window $L$ to 4 weeks. We select counties using the FIPS codes. The final number of counties is 2299. We only exclude counties that have zero cases for all timesteps. Note that our experiment scale is much larger and more granular than existing works[4,5], which allows for a wider scope of possible applications.

We use the mean square error (MSE), mean absolute error (MAE), the coefficient of determination ($R^2$), and the average concordance correlation coefficient (CCC) to evaluate model prediction. The CCC

and $R^2$ are computed as:

$$R^2 = 1 - \frac{\sum (\hat{y}_i - y_i)^2}{\sum (\hat{y}_i - \mu_y)^2} \qquad (1)$$

$$CCC = \frac{2\rho\sigma_{\hat{y}}\sigma_y}{\sigma_{\hat{y}}^2 + \sigma_y^2 + (\mu_{\hat{y}} - \mu_y)^2} \qquad (2)$$

where $\mu_{\hat{y}}$ and $\mu_y$ are the means for the predictions and ground truth, and $\sigma_{\hat{y}}^2$ and $\sigma_y^2$ are the corresponding variances. $\rho$ is the correlation coefficient between the two variables $\hat{y}$ and $y$. Note that the range of $R^2$ is $(-\infty, 1)$, so an extreme value may significantly affect the average value. In contrast, the range of CCC is $(-1, 1)$, so this will be less affected by extreme values.

We split the sequence into train, validation, and test sets in a 3:1:1 ratio. The training sequences are from Sep 04, 2020, to Sep 02, 2021; the validation sequences are from Sep 03, 2021, to Dec 23, 2021, and the test sequences are from Dec 24, 2021, to Apr 15, 2022. We train the model on the training set and save the model and hyper-parameters with the best performance on the validation set. We then test the model on the testing set and report the performance. We train all models five times with different random initializations and calculate the standard deviations. Due to diverse location characteristics, the average hospitalization cases vary from a few to tens of thousands of cases. This large variation in case numbers poses a challenge for the deep learning model to learn stable parameters. We therefore conduct the log transformation on the prediction targets and scale the model predictions back to the original scale, then calculate the performance metrics. We also provide the prediction uncertainty at 90% confidence in all county-level and state-level prediction plots using the conformal methods[20]. The conformal algorithms possess explicit and non-asymptotic guarantees without distributional assumptions or model assumptions so that they can be easily applied to all trained models. The prediction interval $\alpha$ is estimated as:

$$\alpha = [\hat{y} - s\sigma, \hat{y} + s\sigma] \qquad (3)$$

where $\hat{y}$ is the average prediction using the five models with different random initializations, $\sigma$ denotes the standard deviation of five models. The value of parameter $s$ depends on the required confidence. For example, if we aim to obtain the prediction interval with 90% confidence, we calculate the $s$ to make the prediction interval $a$ covers 90% ground truth on the validation set. Then we applied the $s$ on the test set to calculate the test prediction interval. We provide a more detailed model uncertainty analysis in Supplementary Fig. 4.

We use Python 3.9, PyTorch 1.12[21], scikit-learn 1.2, and NumPy 1.19 to collect the data and implement the models. We use the mini-batch gradient descent strategy to train the models, and the batch size is set to 128. We use the Adam optimizer with a learning rate of 0.01 for 300 epochs. We save the model with the highest score on the validation set and report the prediction performances on the testing set. All the experiments are done on the server with Intel i9-13900K CPU, 64 GB RAM, and one NVIDIA RTX 4090 GPU. The HOIST source code is publicly available on GitHub (https://github.com/v1xerunt/HOIST).

### Baseline models
We compare HOIST with the following epidemiology and deep-learning methods.
1. DELPHI-SEIR[22]: This is a variant of the SEIR (susceptible, exposed, infectious, and recovered) epidemiology model. Compared with the traditional SEIR model, DELPHI-SEIR can model hospitalization trends and policy strategies. We use the trust region[23] optimization strategy to estimate the model's parameters.
2. GRU: We input all features into a gated recurrent unit (GRU) model and predict the future number of cases. The GRU model is a variant of RNN, widely applied in multiple pandemic prediction works[24–26]. The hidden dimension of GRU is set to 128.
3. LSTM: We input all features into a long short-term memory (LSTM) model and predict the future number of cases. The LSTM model is another variant of RNN. The hidden dimension of LSTM is also set to 128.
4. ColaGNN[9]: ColaGNN learns the location graph with sequential data to learn spatial relationships for pandemic progression. The hidden dimension of RNN is set to 128, and the convolution filter dimension is set to 64.
5. ACTS[8]: ACTS is a COVID-19 forecasting model which uses the inter-series attention mechanism to learn spatial relationships between locations. The convolution filter dimension is set to 64, and the segment length is set to 14.
6. CovidGNN[27]: CovidGNN uses a graph neural network with skip connections to predict future COVID-19 cases. We use a two-layer graph attention network, and the graph network dimension is set to 64.
7. STAN[4]: STAN fuses the SIR dynamics into a spatiotemporal prediction model for COVID-19 case prediction. The graph network dimension is set to 64, and the hidden dimension of GRU is set to 128. Since the SIR dynamics do not apply to the hospitalization prediction task, we remove the SIR constraints in the STAN model.

All models can access the same data sequences with the same input window and are evaluated on the same testing set. For the spatiotemporal prediction models (i.e., ColaGNN, ACTS, CovidGNN, STAN), we use the static data to build the location graph or calculate the location similarities in their algorithms. For the GRU and LSTM models, we concatenate the static data with the original inputs at each timestep to the model. The SEIR model cannot take the static location data as inputs. All model hyper-parameters are decided by using grid search on the validation dataset. For the DELPHI-SEIR model, we use the deployed version of the DELPHI-SEIR model (i.e., DELPHI-SEIR V4) and recommend optimal parameters. To assess the performance improvement from the Ising dynamics, the adaptative connectivity learning, and the real-world evidence, we conduct an ablation study by comparing HOIST against the following ablation versions from both a data perspective and a method perspective:
1. HOIST-Vaccination: We remove the vaccination statistics data from the model input sequences of HOIST.
2. HOIST-Risk: We remove the high-risk patient statistics in the real-world claims data from model input sequences of HOIST.
3. HOIST-AC: We remove the adaptative connection learning module from HOIST. The **S** matrix is learned by calculating the sequence similarities instead of using location background data.
4. HOIST-Ising: We remove the Ising dynamic loss and replace the EF modeling module with a naïve LSTM network in HOIST.

### Model performance analysis
We design experiments to answer the following research questions:
1. How well does HOIST perform in the hospitalization prediction task?
2. How well does HOIST perform with different lengths of prediction window?
3. How well does HOIST perform under temporal data split setting?
4. What is the analysis of the learned external field weights?
5. How does the HOIST model help to increase vaccination rates effectively?

**28-day hospitalization prediction performance.** Table 1 shows the performance of the 28-day hospitalization prediction task. Compared to the best baseline model, HOIST reaches 70% lower MSE and 50%

**Table 1 | Performance of 28-day hospitalization prediction**

| Model | MSE (×10⁵) | MAE | $R^2$ | CCC |
|---|---|---|---|---|
| DELPHI-SEIR | 400.3 (–) | 1647 (–) | <−1 | 0.02 (–) |
| GRU | 149.2 (6.6) | 897.7 (20.9) | −0.04 (0.04) | 0.27 (0.20) |
| LSTM | 133.9 (6.9) | 826.6 (30.0) | 0.06 (0.04) | 0.48 (0.10) |
| ColaGNN | 138.5 (9.5) | 801.9 (32.1) | 0.10 (0.13) | 0.52 (0.09) |
| ACTS | 127.9 (14.1) | 732.1 (45.3) | 0.19 (0.15) | 0.60 (0.10) |
| CovidGNN | 181.0 (46.2) | 1048.0 (126.2) | −0.26 (0.18) | 0.18 (0.16) |
| STAN | 148.3 (30.7) | 948.2 (68.4) | −0.06 (0.46) | 0.18 (0.12) |
| HOIST-Risk | 65.2 (14.7) | 530.6 (59.5) | 0.41 (0.21) | 0.71 (0.08) |
| HOIST-Vaccination | 62.1 (13.2) | 515.3 (42.3) | 0.45 (0.24) | 0.78 (0.07) |
| HOIST-AC | 73.4 (38.4) | 562.5 (87.9) | 0.27 (0.30) | 0.68 (0.15) |
| HOIST-Ising | 135.0 (15.4) | 735.5 (28.6) | 0.17 (0.14) | 0.39 (0.07) |
| HOIST | **38.5 (10.2)\*** | **368.7 (18.7)\*** | **0.60 (0.16)\*** | **0.89 (0.02)\*** |
| p-value | 3e-6 | 2e-7 | 1e-5 | 7e-4 |

The performance numbers are mean (std). The bold values denote the best results. The asterisk * denotes the performance differences between HOIST and the best baseline models (ACTS) are significant based on the two-sided t-test results ($p < 0.001$). Source data are provided as a Source Data file.

lower MAE. It also achieves an $R^2$ score of 0.6 and a CCC of 0.89, while the best baseline model achieves 0.16 $R^2$ and 0.6 CCC. We find ColaGNN and ACTS achieve better prediction performance among the compared models; this may be because both CovidGNN and STAN require a fixed location graph structure as input, while ColaGNN and ACTS can learn the spatial relationships based on the sequence similarity. Compared to the infected case prediction task, the spatial patterns of hospitalization cases may be more complex and thus cannot be pre-defined using a fixed location graph. Therefore, models that can learn flexible connections commonly outperform all others in this task. By utilizing the location-static background data, HOIST can better extract spatial patterns of the pandemic progression. We also find that traditional epidemiology SEIR models fail to predict accurately in this task, probably because the hospitalization curves do not follow a simple increasing-decreasing trend, which is the underlying assumption of most epidemiological models. Additionally, the ground-truth curves may have multiple peaks and complex short-term variations, which increases the difficulty of the predictive task. Note that when calculating $R^2$ and CCC scores, the average value $\mu_y$ used in the denominator is the average value in the test time phase, which is often difficult to beat since the future $\mu_y$ is unavailable at the prediction time. Therefore, in all experiments, any positive $R^2$ or high CCC scores can be impressive to achieve.

We conduct the Student's t-test to evaluate the significance of the performance differences. The results show that the performance differences between HOIST and the best baseline models are significant ($p < 0.001$). HOIST also outperforms all ablation versions of HOIST by a large margin, validating the effectiveness and necessity of all proposed modules. According to the performance analysis, the Ising dynamics are the most critical component of HOIST. By modeling various influence factors as external fields and using spatiotemporal dynamics to regularize the model learning process, HOIST confirms that the Ising dynamics can improve spatiotemporal predictive performance by more closely resembling the real-world curve. Furthermore, integrating real-world evidence data, such as high-risk patient cohorts and vaccination statistics, can reduce prediction error. Using background data to learn adaptive spatial connectivity in HOIST also enables it to outperform the reduced model, which only uses sequence similarities (i.e., HOIST-AC). The adaptive connectivity learning module also helps the HOIST learn clustered spatial embeddings using the demographics data. We provide visualization plots of the learned

embeddings in Supplementary Fig. 2. The results show that HOIST can not only extract geographical similarities between locations but also can identify geographically distant but socio-economically similar locations. We also provide the predicted curve plots for all US states in Supplementary Fig. 1.

**Prediction performance with different lengths of prediction window.** In this section, we evaluate each model's long-term and short-term prediction performance by changing the prediction window $L$ from 1 to 5 weeks and then train and evaluate all the models with the same strategy. The MAE, MSE, $R^2$, and CCC of HOIST and the 6 best baseline models are shown in Fig. 2. Due to its poor performance, we exclude the SEIR model.

The results demonstrate that the MSE and MAE of all models increase as the length of the prediction window increases. This is expected since predicting further into the future becomes progressively more challenging. However, HOIST consistently achieves a much lower MAE and MSE than all baseline models. HOIST also achieves consistently high $R^2$ and CCC scores. Compared to the baseline models, HOIST achieves, on average, a 48% lower MAE, 65% lower MSE, 272% higher $R^2$, and 51% higher CCC. The results demonstrate that the HOIST model accurately predicts both long-term trends and short-term variations, enabling broader real-world applications.

**Prediction performance under temporal data split setting.** New variants of the COVID-19 virus can lead to different disease severity. It is non-trivial for a model to accurately predict future hospitalization under such distribution transitions. We design experiments to evaluate how model performance evolves over time. We use a sliding window training setting by using 10 weeks data for training and using the next 4 weeks data for testing. We split the time from Sep 2020 to April 2022 into seven periods, and the model is tested in seven testing time phases. We compare HOIST against the other three best baseline models (i.e., LSTM, ACTS, and ColaGNN). The results are shown in Fig. 3.

The results show that HOIST consistently outperforms all baseline models in terms of MSE and MAE in all testing phases. HOIST consistently achieves low prediction errors and high CCC scores and is less affected by temporal data distribution shifts. We notice that some baseline models experience low prediction performance on Feb 2021 and Jan 2022. This may be due to distribution shifts between training and testing data caused by emerging of new variants of the COVID-19 virus. However, HOIST achieves lower prediction errors than baselines on these time phases.

**Analysis of learned external fields.** Compared with the baseline black-box models, modeling external fields with Ising dynamics allows us to analyze the importance of different factors for different locations, providing us with more insight into which features are most predictive of hospitalization. In the learned EF parameters $\hat{\mathbf{f}}$, each dimension denotes the weight of the corresponding input factor. To visualize the effect of the number of vaccinations on the hospitalization rate, we plot the corresponding weights of the following vaccination features: (1) the total number of vaccinations, (2) the number of first vaccination administrations, (3) the number of second vaccination administrations, and (4) the number of booster administrations, as shown in Fig. 4.

The negative EF weights occur because the vaccination ratio is negatively correlated with the hospitalization case count. However, the results show that the EF weights of all four features are increasing, meaning the predictiveness of vaccinations is decreasing over time. This is possibly due to the increased prevalence of new variants of the virus, many of which are more resistant to vaccinations, which is consistent with conclusions in recent medical research[28,29]. However, we can still observe that the booster doses (the purple line) are more

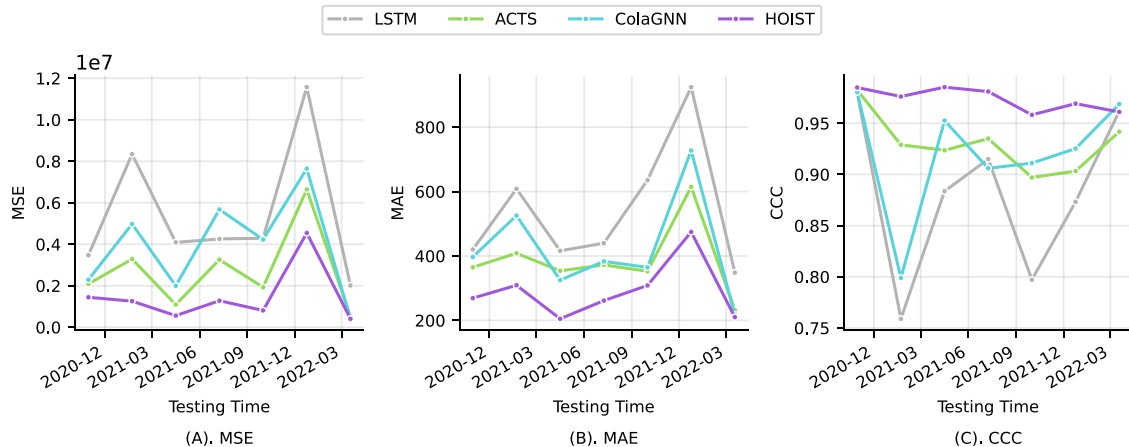

**Fig. 2 | Prediction performance plots under different lengths of prediction window _L_.** Data are presented as mean values ± standard deviations. The error bars are standard deviations over five experiments with random initializations with _n_ = 2299 locations. Source data are provided as a Source Data file.

**Fig. 3 | Prediction performance under different time split in 3 panels.** Each dot denotes a testing phase. Source data are provided as a Source Data file. **A** Mean square error (MSE), **B** mean average error (MAE), and **C** concordance correlation coefficient (CCC).

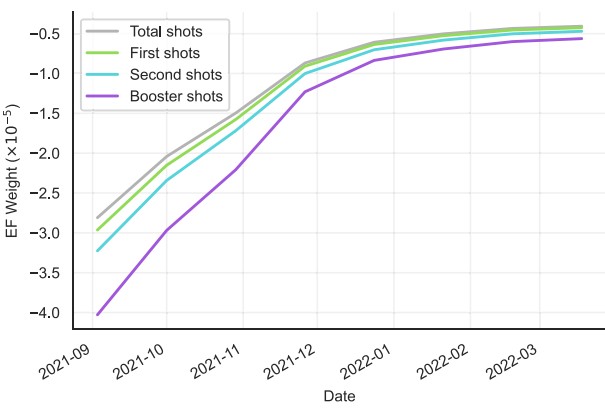

**Fig. 4 | Temporal change of external field weights for vaccination features.** Source data are provided as a Source Data file.

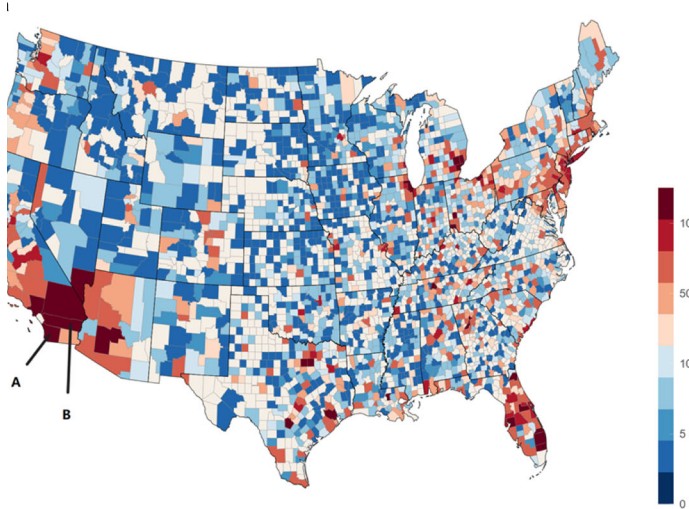

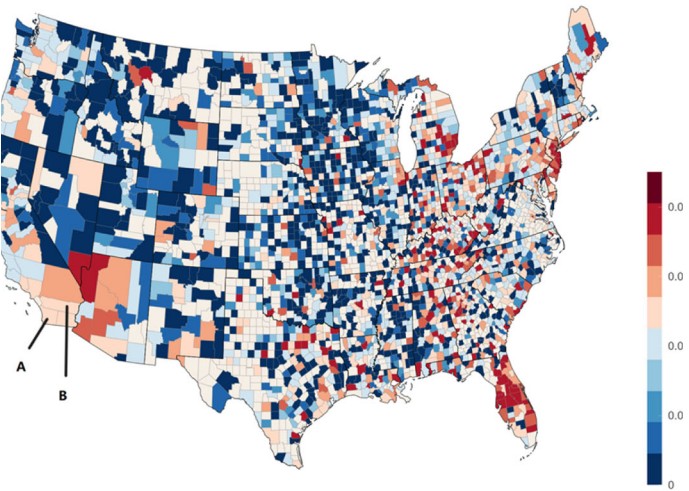

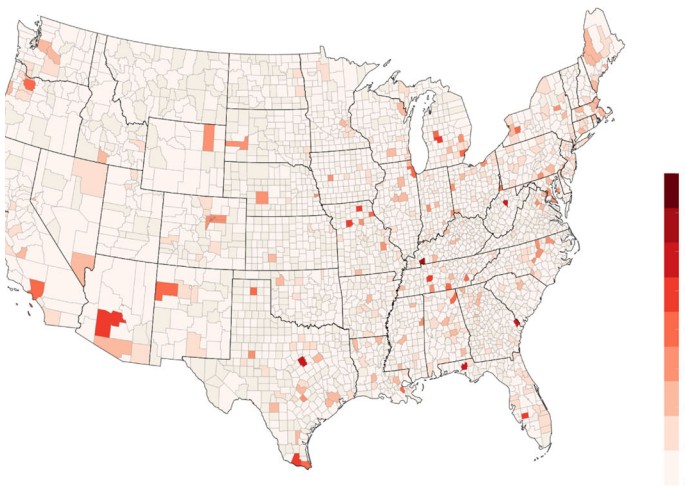

effective than the first and second doses ($p < 0.05$). We also compare the model effect (i.e., the EF weights) of Pfizer (CPT code 0001A ~ 0004A) and Moderna (CPT code 0011A ~ 0013A, 0064A) COVID-19 vaccines, and we find that their effects do not have a statistical difference in our model ($p$-value 0.1).

We also explore the EF weights of high-risk conditions, which are condition codes defined by the CDC COVID-19 People with Certain Medical Conditions guideline[14] and the Charlson Comorbidity Index[15]. These features are proven to have significant impacts on COVID-19 patients' outcomes[30,31]. We find the feature with the highest EF weight is the number of patients older than 65. The top five conditions are renal disease, dementia, immunodeficiency, malignancy, and chronic lung disease. The results are consistent with CDC guidelines, and we do not observe statistically significant differences in these five features.

**Predicted hospitalization hotspot map and error analysis.** We present the predicted hospitalization hotspot map in Fig. 5. Figure 5a is the hotspot map colored by the number of hospitalization cases and Fig. 5b is colored by the ratio of hospitalization cases over the total population. There are some blank areas in the map in cases of no valid data or zero case counts in these counties. These heatmaps can be accessed in an online visualization platform at https://v1xerunt.github.io/HOIST/.

We find that the spatial patterns of hospitalization are similar in both figures. West coast, Northeast coast, and Midwest areas generally have high hospitalization case counts and per-person occurrence. Interestingly, we also find that some locations—such as the San Diego county (A) and Riverside county (B) along the West coast—have very high case counts but only medium-level hospitalization ratios. A possible interpretation could be that the people from these counties tend to travel to Los Angeles for hospitalization.

We also plot a heatmap to show the spatial patterns of the mean average percentage error (MAPE). We only calculate the MAPE for locations where case counts are larger than 100 in the testing phase because a small prediction error can lead to very large MAPE for locations with low counts. The results are shown in Fig. 5c.

We find that HOIST achieves low MAPE for most locations, with only 15 locations having a MAPE greater than 0.5. For these locations, HOIST tends to underestimate the case counts. This may be because there are some pattern changes in the testing time window; for example, in Effingham, Georgia, HOIST has 0.64 MAPE. When we plot the predicted hospitalization curve and the number of infected cases (Fig. 6), we find several peaks in the infected case count curve during both the validation and testing phases, which is quite different from training patterns. These sudden fluctuations in input features may cause HOIST to fail to accurately predict the surge of hospitalization

cases during the testing phase. We provide another two case examples for error analysis in Supplementary Figure 3.

**Number needed to treat prediction with HOIST.** Increasing a community's vaccination rate is an effective method of reducing both the occurrence and severity of COVID-19[32]. However, there is still uncertainty surrounding how and whether this effectiveness is influenced by

**Fig. 5 | Predicted hospitalization hotspot and error map by HOIST. a** Predicted hospitalization hotspot map in future 28 days by HOIST, colored by case count. West coast, Northeast coast, and Midwest areas generally have high hospitalization case counts. **b** Predicted hospitalization hotspot map in the future 28 days by HOIST, colored by hospitalization ratio. We find two example locations, San Diego county (A) and Riverside county (B), along the West coast, have very high case counts but only medium-level hospitalization ratios. **c** Prediction errors for locations with more than 100 cases in the testing time window, colored by mean average percentage error. HOIST achieves low prediction errors in most locations. Source data are provided as a Source Data file.

local vaccination rates. In other words, we ask the question, "Due to the disproportionately higher vaccination rates in urban areas, do inequalities exist regarding the number of additional vaccinations necessary to reduce a single hospitalization or death?" We also calculate the cost-effectiveness of future vaccination efforts based on the county-level average cost of COVID-19 hospitalization. Accurately answering these questions at the county level may allow policymakers and healthcare institutions with limited distribution capabilities and finite medical resources to develop more targeted vaccination efforts.

The high predictive accuracy of HOIST enables us to answer these questions. By changing the vaccination rates in the input features, HOIST can simulate the changes in the predicted hospitalization curve. We use the predicted number of hospitalizations to calculate the number needed to treat (NNT) with respect to how many vaccinations are needed to prevent one hospitalization in each county. For each location, we increase the vaccination rate by 10% (i.e., 10% total population) and calculate the predicted hospitalization case reduction between the original prediction $\hat{y}$ and the simulated prediction $\hat{y}_{sim}$. The NNT for hospitalization is calculated as:

$$\text{NNT}_h = \frac{1}{\text{CER} - \text{EER}} = \frac{1}{\left(\frac{\hat{y}}{0.1p} - \frac{\hat{y}_{sim}}{0.1p}\right)} = \frac{p}{10(\hat{y} - \hat{y}_{sim})} \quad (4)$$

where CER denotes the control event rate (original predicted hospitalization rate), EER denotes the experimental event rate (simulated hospitalization rate after the vaccination rate increases by 10%), and $p$ denotes the population size of the location.

Furthermore, we acknowledge that morbidity and mortality among individuals hospitalized with COVID-19 occur disproportionately in certain ethnic and racial minority groups. To address these outcome disparities in our model, we adjust the NNT for each county based on its demographics and their respective risk ratios:

$$\text{NNT}_a = \frac{\text{NNT}_h}{1*r_{\text{white,nh}} + 0.8*r_{\text{asian,nh}} + 1.7*r_{\text{black,nh}} + 1.8*r_{\text{hispanic}}} \quad (5)$$

where $r_{\text{white,nh}}$, $r_{\text{asian,nh}}$ and $r_{\text{black,nh}}$ denote the population ratio of White, Asian, and Black non-Hispanic persons, and $r_{\text{hispanic}}$ denotes the population ratio of Hispanic or Latino persons. All county-level population ratios are extracted from the census data. The risk adjustment factor for each race and ethnicity is based on national COVID-19 death risk ratios reported by the CDC[33,34]. While $\text{NNT}_h$ represents the predicted number of vaccinations needed to prevent one hospitalization, $\text{NNT}_a$ is not a direction calculation of NNT for death prevention; instead, its purpose is to illustrate how locations with similar $\text{NNT}_h$ might be further stratified based on the predicted outcomes of those predicted hospitalizations. The resultant $\text{NNT}_a$ favors locations with more Black/African American, Hispanic, and Latino populations. Future analytical and predictive efforts incorporating county-level death rates are warranted.

The heatmaps of $\text{NNT}_h$ and $\text{NNT}_a$ are shown in Fig. 7. Note how the count and ratio maps identify hotspots primarily among urban and metropolitan areas. In contrast, the NNT values are generally lowest in the rural region, which are highly clustered in certain states, including

North Dakota, South Dakota, Kansas, Nebraska, Montana, etc. Quantitatively, increasing the vaccination ratio by 10% can reduce the number of current hospitalization cases by 15% on average for all locations.

We further calculate the cost ratio between state average vaccination costs and state average COVID-19 hospitalization costs. We collect the state-level average vaccine cost from the Centers for Medicare and Medicaid Services (CMS)[35] and hospitalization cost for both complex and noncomplex COVID-19 inpatients from the COVID-19 Cost Tracker[36]. The cost ratio is then calculated as:

$$r_{\text{cost}} = \frac{r_{\text{complex}}*c_{\text{complex}} + r_{\text{noncomplex}}*c_{\text{noncomplex}}}{\text{NNT}_h*c_{\text{shot}}} \quad (6)$$

where $r_{\text{complex}}$ and $r_{\text{noncomplex}}$ are the average complex and noncomplex COVID-19 inpatients ratio extracted from the hospital resource data. $c_{\text{complex}}$ and $c_{\text{noncomplex}}$ are hospitalization costs for complex and noncomplex COVID-19 inpatients. $c_{\text{shot}}$ is the cost of one vaccine. All parameters to calculate the $r_{\text{cost}}$ vary by location.

The cost ratio heatmap is shown in Fig. 7c. A cost ratio greater than 1 indicates that increasing the number of vaccinations in that location by the NNT is less expensive than the cost of a single prevented hospitalization. The heatmap demonstrates a ratio favoring future vaccination efforts in primarily rural regions, most commonly among centrally located and landlocked counties. We list the $\text{NNT}_h$ and $\text{NNT}_a$ of the top 15 locations with the highest (most favorable) cost ratios in Table 2. The full county-level table can be accessed in the GitHub repository and the visualization platform. These results suggest three meaningful conclusions. First, even small-scale vaccination efforts in targeted counties are likely to prevent at least one hospitalization and its related sequelae. Second, rural vaccination outreach efforts are likely cost-saving endeavors in 368 counties among 43 states (with a cost ratio >1). Third, we find that locations that have high cost ratio also have high adjust ratio. This may indicate large healthcare disparities for different races in these locations. More life could be saved by improving vaccination fairness in these locations.

## Discussion

In this work, we propose an Ising dynamics-based deep learning model, HOIST, for spatiotemporal COVID-19 hospitalization prediction. The HOIST model is built with multiple data sources, including disease and vaccination statistics from real-world medical claims, medical resource usage, census, and geographical and mobility data. HOIST can handle complex inner-region influence factors and adaptively learn the inter-regional relationships without requiring a predefined location graph with fixed edges. By drawing the analogy between locations and lattice sites, we use the Ising dynamics as a regularizer to guide the model learning and prediction process, which is a natural choice to model the inner-region factors and inter-region similarities simultaneously.

We evaluate the HOIST model on the large-scale spatiotemporal COVID-19 hospitalization prediction task for 2299 counties in the United States. In the 4-week hospitalization prediction task, HOIST achieves 368.7 MAE, 0.6 $R^2$ and 0.89 CCC scores. For different lengths of prediction window from 1 to 5 weeks, HOIST achieves 48% lower MAE, 65% lower MSE, 272% higher $R^2$ and 51% higher CCC on average compared to the best baseline models. We conducted a detailed analysis of the model results and learned parameters. We find that the booster shot of vaccination population percentage has a more significant negative correlation to future hospitalization cases than the first and second vaccination shots. We also find that the effects of two major vaccination brands, Pfizer and Moderna, have no statistical difference in our model.

We note that, in contrast to many previous models, our HOIST-based clinical and economic predictions suggest a need to prioritize

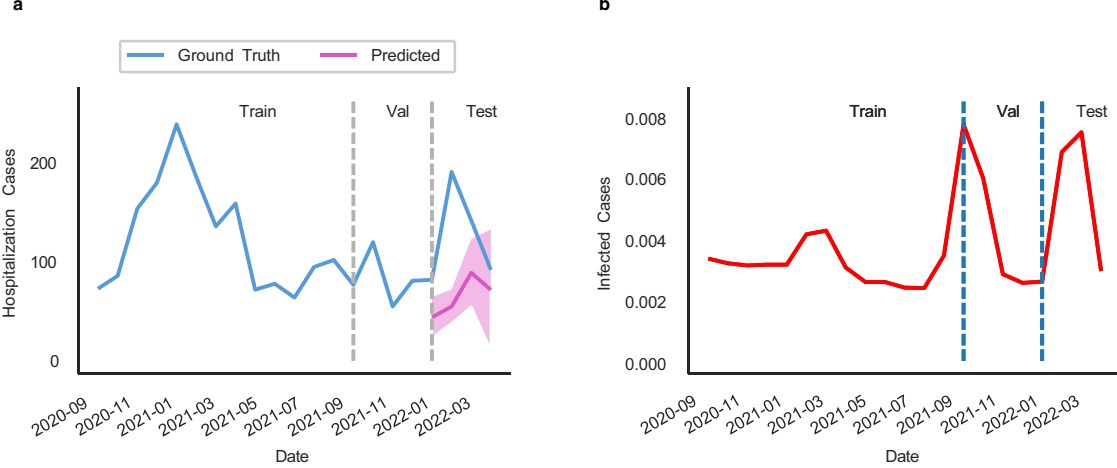

**Fig. 6 | Predicted hospitalization and infected case count curves for Effingham, Georgia. a** Predicted curve by HOIST. The line denotes the mean value, and the shadowed area denotes the prediction interval with 90% confidence. **b** Curve of infected case count. Source data are provided as a Source Data file.

future vaccination efforts in rural areas over urban centers. The results indicate that increasing the total vaccination ratio by 10% can make the number of current hospitalization cases reduce by 15% on average for all locations. We believe these predictions accurately reflect the growing disparity in vaccination rates between these communities. Additionally, we note a positive correlation between vaccination rate and NNT; in fact, the adjusted NNT map appears strikingly similar to maps depicting percentages of fully vaccinated residents by county[37]. We surmise that communities with lower-than-average proportions of vaccinated individuals are more likely to benefit from further vaccination efforts, irrespective of rurality. For 368 counties among 43 states, increasing the vaccination ratio in these counties can significantly reduce the overall healthcare costs for COVID-19 patients. We hope these results and our model can inform clinicians, healthcare institutions, and policymakers to improve their decisions and ultimately reduce the negative economic and health impacts caused by the pandemic.

This research is not without limitations. In the data collection process, the number of hospitalization cases may have bias. We define a hospitalization case as a patient hospitalized within 35 days after the COVID-19 diagnosis. We extract these statistics from the claim dataset, so we cannot know if COVID-19 is the primary cause of hospitalization. Though this definition is the same as existing COVID-19 prediction works[18,38], the model may conflate hospitalization trends due to other causes with those accurately attributed to COVID-19. This is especially true in the pediatric population, where COVID-19 infectivity is high but severe cases are uncommon. Despite these limitations, we still select claims data as our primary data source as it provides valuable granular county-level statistics.

The second limitation concerns data preprocessing. The original data distribution of hospitalization cases is highly skewed. Our dataset is much larger than previous research; over 80% of locations have fewer than 1000 hospitalization cases in the 28-day prediction window, and only 2% have more than 10,000 hospitalization cases. To maintain the stability of the machine learning models, we conduct a log transformation on the prediction target, followed by a *z*-score transformation to normalize the sequence. Therefore, model prediction results require two reverse transformations to return to the original scale. This transformation may cause a small error in the prediction scale to grow exponentially as it converts to the original scale. As a result, we observe that baseline models sometimes have abnormally large prediction errors and standard deviations (e.g., over 10,000 MAE), especially when the prediction window is long. We manually remove these outliers from the performance table. Though

we do not find these issues in HOIST, which further demonstrates the stability of the model, we still believe this is an open research challenge in the large-scale spatiotemporal prediction work. The data scale issue might be solved by proposing more advanced preprocessing techniques or new scale-invariant models.

A third limitation exists at the methodology level. Though the spatial connectivity learning module is more flexible than previous works requiring fixed graph structures, the learned connectivity does not change with time. In real-world scenarios, spatial connectivity may be affected by several factors, including weather, holidays, and travel restrictions. Therefore, future works may include integrating more data sources, such as real-time mobility data between locations, to further improve the extracted spatial patterns, interpretability analysis on the feature contributions, and more granular age information. Besides, the Ising dynamics maybe not be the only choice to model the spatial and social background factors for hospitalization prediction. The reason we choose the Ising dynamics is that it can incorporate both inter-region spatial relationships and inner-region factors and can handle these complex factors compared to naïve SIR and SEIR models. We are inspired by previous sociology studies that use Ising models for human behavior tendencies[39]. Though the Ising dynamics show good prediction performances in HOIST, we hope our exploratory work can provide more inspiration for future studies using other real-world dynamics.

## Methods
We follow the recommendations set out in the Global Code of Conduct for Research in Resource-Poor Settings when designing, executing, and reporting the research, and this research does not use individual-level data. Our study complies with the recommendations of the GATHER statement.

### Background
Machine learning and deep learning models have been widely applied in pandemic predictions. Statistical epidemic prediction models, including SIR, SEIR, and their variants, have achieved some success in infection, hospitalization, and mortality prediction tasks[25,40,41]. To extract complex temporal patterns, some works have applied the recurrent neural network (RNN) and its variants, such as long short-term memory network (LSTM) and gated recurrent unit network (GRU), to predict future infected and hospitalization cases[24–26].

To further improve the spatiotemporal prediction performance, a significant line of research focuses on extracting and utilizing spatial dependencies. Graph neural networks (GNN)[4,27,42,43] and

a

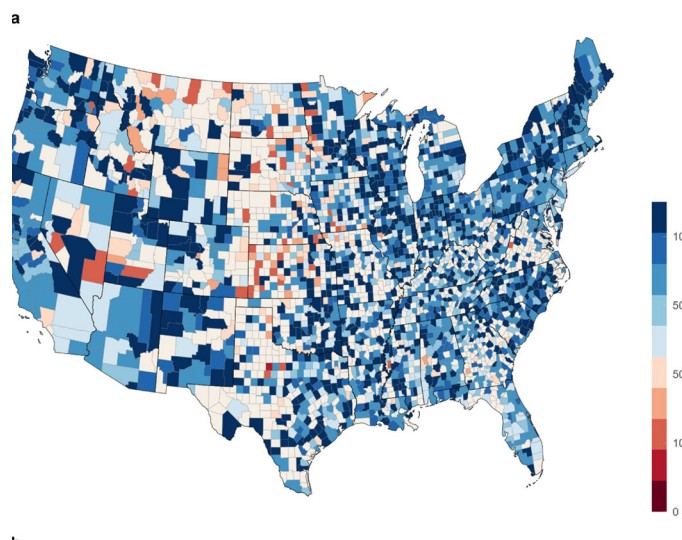

b

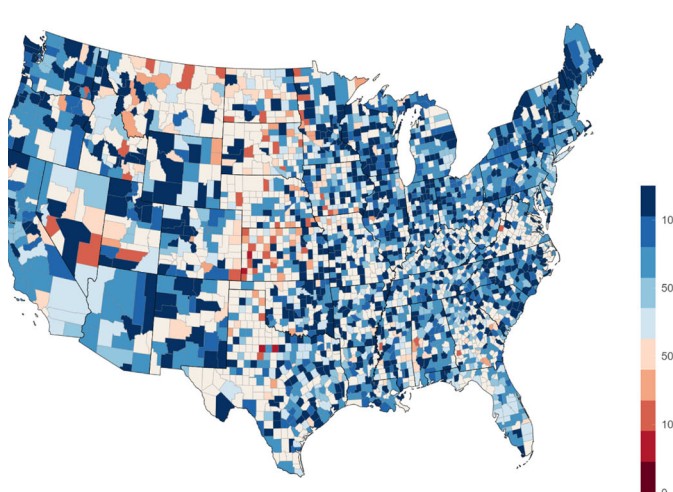

c

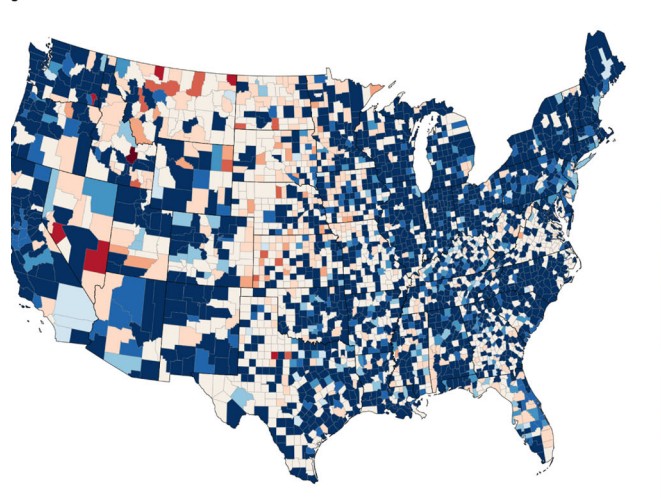

**Fig. 7 | County-level number needed to treat (NNT) and cost ratio heatmap. a** Predicted number needed to treat (NNT) heatmap by HOIST, colored by $NNT_h$. Red locations indicate that only a few vaccine shots can help reduce a hospitalization case. **b** Race-adjusted number needed to treat (NNT) heatmap by HOIST, colored by $NNT_a$. The spatial pattern is similar to the unadjusted map, but some locations are affected by the race percentage. **c** Heatmap of cost ratio between hospitalization costs and vaccination costs. Red locations indicate that they can save healthcare costs by giving more vaccinations (i.e., cost ratio >1). Source data are provided as a Source Data file.

**Table 2 | Number needed to treat (NNT) for hospitalization and death of the top 15 locations ranked by the cost ratio**

| County | State | Hospitalization NNT | Adjusted NNT | Cost ratio |
|---|---|---|---|---|
| Butte | Idaho | 130 | 73 | 60.3 |
| Woodbury | Iowa | 76 | 44 | 34.5 |
| Toole | Montana | 230 | 130 | 34.5 |
| Sheridan | Montana | 204 | 115 | 34.0 |
| Broadwater | Montana | 186 | 109 | 33.7 |
| Mineral | Nevada | 273 | 159 | 32.1 |
| Garfield | Washington | 156 | 89 | 31.8 |
| Stonewall | Texas | 71 | 42 | 31.8 |
| Lincoln | Nevada | 201 | 114 | 31.2 |
| Stanton | Kansas | 101 | 67 | 29.2 |
| Oneida | Idaho | 287 | 163 | 28.7 |
| Bowman | North Dakota | 142 | 81 | 28.1 |
| Phillips | Montana | 234 | 132 | 27.4 |
| Ness | Kansas | 145 | 84 | 25.2 |
| Teton | Montana | 308 | 172 | 25.0 |

Source data are provided as a Source Data file.

These manually defined location graphs may differ greatly from the ground-truth inter-regional relationships and spatial pandemic transmission patterns. Besides, some metapopulation models are only applicable for infectious diseases since the underlying assumption is the population flow, which makes it difficult to extract other spatial patterns such as hospitalizations. Some works aim to solve this issue by adaptively learning the connection weights between locations. Jin et al.[8] and Deng et al.[9] utilize the attention mechanism to predict future infected cases in pandemics. Their work learns the connection weights based on the similarities of historical case curves at different locations but largely ignores the underlying social, economic, medical, and geographical similarities, which may have a profound impact on the interlocation relationships. Therefore, further exploration is needed to discern how best to model spatial dependencies with multi-source background data in a flexible way.

Another line of research focuses on integrating real-world disease transmission dynamics into deep learning models. Though deep learning networks can extract complex temporal patterns, they can only predict known data patterns and thus have worse long-term prediction accuracy. Integrating epidemiology models such as SIR and SEIR can help deep learning models predict curves resembling real-world transmission patterns. Gao et al.[4] used an SIR model as a regularizer term in the loss function to help the GAT-GRU model predict county-level COVID-19 infection case numbers in the United States. Storlie et al.[6] proposed a stochastic SIR model to predict future COVID-19 cases. Though these models can be applied in the COVID-19 hospitalization prediction, their main approach involves dividing the infected population into hospitalized and non-hospitalized cohorts and then modifying the transmission equation accordingly. However,

metapopulation analysis models[44–46] have achieved remarkable success in solving this issue. In these works, counties and states are modeled as nodes in the graph, and the edges are defined using geographical and sociological similarities or mobility scores. By combining the GNNs and RNNs, these models can extract spatio-temporal patterns in the data and make better predictions. However, most of these models rely on pre-defined location graphs.

as mentioned in the introduction, the relatively few parameters in most epidemiological models may not fully describe the influences of complex underlying disease distribution, viral variants, and vaccination effects. Additionally, these epidemiological models are not designed for general spatiotemporal predictions and thus cannot extract and utilize inter-regional interactions. Therefore, integrating appropriate real-world dynamics that can naturally model spatiotemporal hospitalizations may improve the prediction performance. The Ising model is a statistical mechanics model for site spin configuration estimations; it has been applied in sociology research to model social behavior by considering each individual as a site[39]. In this work, instead of directly using statistical methods to estimate the Ising parameters in a low-dimensional and static-variable-only dataset, we combine the Ising dynamics with deep learning techniques to model the spatiotemporal pandemic hospitalization patterns from complex, high-dimensional multi-source data.

### Learning adaptive connectivity in the latent space

Pandemic progression patterns have spatial similarity depending on the underlying population, economic, geographic, and mobility factors in various locations. Recent works suggest that utilizing these spatial similarities can guide the model to better extract progression patterns and make more accurate predictions[4,27,42,43]. In this work, instead of pre-defining a connected location graph with fixed edges, we aim to learn the adaptive connectivity between locations. Concretely, we measure the latent distances between locations from multiple perspectives from the static data and then normalize the distances to similarities.

In HOIST, each location will be embedded in a multi-dimensional latent space. The latent space is created by considering the set of socio-demographic variables as dimensions so that each location will have a unique position, and locations that are similar in these socio-demographic factors will be embedded closely in the space (also known as the Blau space in sociology study)[47]. To calculate the distance of two locations in the latent space, previous works directly use the sum of Euclidean distance over all dimensions[39]. This is applicable in small-scale settings, but our setting is large-scale and high-dimensional, containing all counties in the United States. The scale differences in some factors are substantial (e.g., population size and geographic distance). Using Euclidean distance may lead to biased connectivity. Furthermore, we also aim to capture the complex nonlinear similarity, which is difficult for scale-invariant distance metrics such as Mahalanobis distance. In this work, we calculate the distances using the graph attention mechanism[48]. Concretely, for location $i$ and $j$, the latent distance $l_{ij}$ is calculated as:

$$l_{ij} = \sum_{\forall \mathbf{K} \in \{\mathbf{M}, \mathbf{E}\}} \sigma\left(\mathbf{W}_{K,2}(\mathbf{W}_{K,1}\mathbf{k}_i \| \mathbf{W}_{K,2}\mathbf{k}_j)\right) + \sigma\left(\mathbf{W}_{D,2}(\mathbf{W}_{D,1}(d_{ij} \| g_{ij}))\right) \quad (7)$$

where $\mathbf{k}_i$ denotes the $i$-th row vector of the $\mathbf{K}$ matrix (i.e., the population demographics $\mathbf{M}$ and the economics and healthcare statistics $\mathbf{E}$), $d_{ij}$ and $g_{ij}$ denote the value at $i$-th row and $j$-th column in the matrices $\mathbf{D}$ and $\mathbf{G}$ (i.e., the distance between location $i$ and $j$), $\mathbf{W}_{K,2}, \mathbf{W}_{K,1}, \mathbf{W}_{D,1}, \mathbf{W}_{D,2}$ are weight matrices, $\sigma$ denotes the sigmoid activation function, and $(\cdot \| \cdot)$ denotes the concatenation operation. Using this attention mechanism, the model can adaptively learn the distance between two locations from multiple perspectives, including demographics, populations, economics, healthcare, geographic distance, and mobility. The distances are further normalized to obtain the similarity $s_{ij}$ as:

$$s_{ij} = \frac{\exp\left(l_{ij}\right)}{\sum_k^N \exp\left(l_{ik}\right)} \quad (8)$$

where $s_{ij}$ indicates the normalized adaptive similarity between location $i$ and $j$ learned by the model. The higher the $s_{ij}$, the more similar the two locations. The similarity scores of all locations formulate the matrix $\mathbf{S} \in \mathbb{R}^{N \times N}$ which is used to inform how the model utilizes the learned spatial connectivity in the following sections.

### Modeling the external fields in Ising dynamics

In the Ising model, the energy of a spin configuration $\mathbf{y}$ of lattice sites is given by the Ising Hamiltonian $I$, which takes the form:

$$I = -\sum_i \left(e_i y_i + \sum_{<i,j>} s_{ij} y_i y_j\right) \quad (9)$$

where $s_{ij}$ denotes the interaction between two sites $i$ and $j$, and $e_i$ denotes the external field (EF) interacting with site $i$. The Ising Hamiltonian $I$ can be considered as the sum of the energy related to the interactions between sites and the energy related to the external field. This Ising Hamiltonian form is applied to capture pandemic hospitalization patterns. First, the hospitalization patterns are highly correlated with the location-specific distributions on comorbidity, vaccination, and hospital resource usage. This can be modeled as the first term in the Ising Hamiltonian $e_i y_i$. Second, the hospitalization patterns are highly correlated across locations depending on the underlying socioeconomic factors. The second term in the Ising Hamiltonian matches location correlation. We propose to use the Ising dynamics to guide the model learning process in this manner.

To fuse this dynamic equation into the HOIST model, the initial step is to model the external fields using Ising dynamics. More specifically, we model the external fields of a specific location from multiple perspectives. The idea is to utilize historical statistics to infer future hospitalization cases. For example, if the number of infected cases among high-risk cohorts has risen in the past few months, the number of hospitalization cases is also likely to rise soon. This temporal delay is expected since many infected patients, even those at high risk for critical illness, do not develop severe manifestations immediately. Of course, we also take vaccinations into consideration; as the number of immunized individuals increases, the hospitalization rate decreases. All sequence vectors are flattened into vectors in a historical window $\tau$ instead of just the values at timestep $t$ (e.g., $\mathbf{u}'_t = [\mathbf{u}_{t-\tau+1}, \mathbf{u}_{t-\tau+2}, \ldots, \mathbf{u}_t]$). Likewise, we create the historical new infected case sequence $\mathbf{i}'_t$, claim statistics sequence $\mathbf{c}'_t$, vaccination statistics $\mathbf{v}'_t$, and medical resource usage sequence $\mathbf{u}'_t$ at timestep $t$. Note that we omit the location index $i$ in this section because we are only discussing one specific location. The parameters of external fields are modeled as:

$$\mathbf{f}_I = \sigma(\phi_I(\mathbf{i}'_t)) \quad (10)$$

$$\mathbf{f}_C = \sigma(\phi_C(\mathbf{c}'_t)) \quad (11)$$

$$\mathbf{f}_V = \sigma(\phi_V(\mathbf{v}'_t)) \quad (12)$$

$$\mathbf{f}_U = \sigma(\phi_U(\mathbf{u}'_t)) \quad (13)$$

where $\phi : \mathbb{R}^F \to \mathbb{R}^F$ denotes the projection function to generate scaling factors for each sequential factor. By multiplying the scaling factors and the sequential factors, the final predictions are calculated as:

$$\hat{\mathbf{f}} = [\mathbf{f}_I \cdot \mathbf{i}'_t \| \mathbf{f}_C \cdot \mathbf{c}'_t \| - \mathbf{f}_V \cdot \mathbf{v}'_t \| \mathbf{f}_U \cdot \mathbf{u}'_t] \quad (14)$$

$$\hat{\mathbf{h}}_t = \text{LSTM}\left(\hat{\mathbf{f}}, \hat{\mathbf{h}}_{t-1}\right) \quad (15)$$

Here LSTM($\cdot$) denotes a multi-layer perceptron to capture temporal progressions, the ($\cdot$) operation denotes the Hadamard product and ($\cdot$||$\cdot$) denotes the concatenation operation. This process is essentially learning weights (i.e., $\mathbf{f}_I$, $\mathbf{f}_C$, $\mathbf{f}_V$ and $\mathbf{f}_U$) for different influence factors. The larger the weight, the greater the influence of the corresponding factors on the prediction result. This is also why we added the negative symbol before $\mathbf{f}_V$, since the vaccinations should be negatively correlated with hospitalization cases. By concatenating all the scaling factors, we obtain the EF strength vector $\hat{\mathbf{f}}$. We further use an LSTM network to model the temporal progression of the EF.

To generate the final predictions for location $i$ (here, we bring back the location subscript as $\hat{\mathbf{h}}_{i,t}$), we calculate the weighted sum of the final hidden state from the LSTM network from other locations. We use the adaptative connectivity $\mathbf{S}$ as the connection weight as:

$$\mathbf{z}_{i,t} = \hat{\mathbf{h}}_{i,t} + \sum_j^N s_{ij}\hat{\mathbf{h}}_{j,t} \tag{16}$$

$$\hat{y}_{i,t} = \text{MLP}(\mathbf{z}_{i,t}) \tag{17}$$

In this way, the final prediction result $\hat{y}_{i,t}$ considers not only the temporal progression of the pandemic and influence factors (Eq. 3) but also the relevant spatial relationships (Eqs. 2 and 4).

### End-to-end learning with the spatiotemporal Ising loss function

Our loss function consists of a prediction loss and an Ising dynamics constraint loss. First, the prediction loss is calculated by using the mean square error:

$$\mathcal{L}_p = \sum_i^N \sum_t^T \left( \hat{y}_{i,t} - y_{i,t} \right)^2 \tag{18}$$

The second loss term is the Ising constraint loss $\mathcal{L}_d$, which takes the form:

$$e_i = \sum_k^{|\hat{f}_i|} \hat{f}_{i,k} \tag{19}$$

$$\hat{y}_{i,t}^d = e_i \hat{y}_{i,t} + \sum_j^N s_{ij} \hat{y}_{i,t} \hat{y}_{j,t} \tag{20}$$

$$\mathcal{L}_d = \sum_i^N \sum_t^T \left( \hat{y}_{i,t}^d - y_{i,t} \right)^2 \tag{21}$$

The term $e_i$ denotes the summation over all dimensions of $\hat{\mathbf{f}}_i$, and $\hat{f}_{i,k}$ denotes the $k$-th dimension in the $\hat{\mathbf{f}}_i$ vector, which turns the EF vector into a scalar, denoting the EF strength. $s_{ij}$ is the value at the $i$-th row, $j$-th column in the $\mathbf{S}$ matrix, which is the similarity score between location $i$ and $j$. $\mathcal{L}_d$ is essentially another mean square error loss between the ground-truth value and the estimated value using the Ising dynamics (i.e., $\hat{y}_{i,t}^d$). In the original Ising dynamics setting, the system's energy should be as low as possible to keep the system stable. Here, we use the same idea to reduce the prediction error $\mathcal{L}_d$ as part of our loss function. It is worth noting that $\hat{y}_{i,t}$ is the final prediction result and $\hat{y}_{i,t}^d$ is only used as an auxiliary output to optimize the Ising constraint parameters. The final loss takes the form:

$$\mathcal{L} = \mathcal{L}_d + \mathcal{L}_p \tag{22}$$

By optimizing the Ising constraint loss and the prediction loss together, the model can extract the temporal hospitalization progression patterns as well as optimize the spatial adaptive connectivity. Using the Ising dynamics as a constraint instead of directly using $\hat{y}^d$ as the prediction result and only optimizing $\mathcal{L}_d$ allows the model to have more flexibility to learn spatiotemporal patterns from high-dimensional data.

### Reporting summary
Further information on research design is available in the Nature Portfolio Reporting Summary linked to this article.

## Data availability
The mobility scores are collected from the Multiscale Dynamic Human Mobility Flow Dataset[11] (https://github.com/GeoDS/COVID19USFlows). The census features are collected from the county-level census dataset provided by the US County-level Dataset[12] (https://github.com/JieYingWu/COVID-19_US_County-level_Summaries). The daily new cases are collected from the Johns Hopkins COVID-19 Data Repository[13] (https://github.com/CSSEGISandData/COVID-19). The medical resource usage statistics are collected from HealthData.gov[17] (https://healthdata.gov/Hospital/COVID-19-Reported-Patient-Impact-and-Hospital-Capa/anag-cw7u). All the processed data are available at https://github.com/v1xerunt/HOIST. Our model can be trained without using claims data. Source data of tables and figures are provided with this paper. Source data are provided with this paper.

## Code availability
The codes for model construction, training, and inference used in this paper are publicly available at https://github.com/v1xerunt/HOIST. The visualization results are available at https://v1xerunt.github.io/HOIST/.

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

## Acknowledgements

This work was supported by NSF awards SCH-2205289, SCH-2014438, IIS-2034479, NIH award R01 1R01NS107291-01, IQVIA, and OSF Healthcare. J.G. acknowledges the receipt of studentship awards from the Health Data Research UK-The Alan Turing Institute Wellcome PhD Programme in Health Data Science (Grant Ref: 218529/Z/19/Z). We thank Zhen Lin from the University of Illinois Urbana Champaign for assisting with the model uncertainty estimation. Icons in the figures are designed using resources from Flaticon.com. For the purpose of open access, the author has applied a Creative Commons Attribution (CC BY) license to any author-accepted manuscript version arising from this submission.

## Author contributions

J.G. developed the method and conducted all the experiments. C.M. and L.G. provided the datasets. J.H. managed the project meetings. A.C. provided the clinical assessment. A.C. and J.S. co-led the overall project. All authors (J.G., J.H., C.M., L.G., A.C., and J.S.) participated in paper writing.

## Competing interests

The authors declare no competing interests.
