## [Peer Review File · Nature Communications]

REVIEWER COMMENTS

Reviewer #1 (Remarks to the Author):

In this paper, the authors developed a deep learning algorithm, HOIST, to predict COVID-19 hospitalization. The major contribution of the algorithm is the introduction of an Ising-type regularization into the loss function. Essentially, this regularization term imposes a constraint on the spatial structure of hospitalization in the latent space. The temporal structure is captured by the LSTM. The training process then optimizes the spatio-temporal structure of hospitalization together. I think this is an interesting study applying deep learning techniques to healthcare. However, I have a few concerns on the methodology and applicability in real-world settings.

1. In statistical physics, the Ising Hamiltonian represents the energy of the system. By varying the state of each site, the goal is to find the configuration with the lowest energy. In HOIST, however, the functional form of the Ising Hamiltonian was used to impose a spatial constraint. The hospitalization in each location cannot be varied arbitrarily and the goal in HOIST is to minimize the distance between the Ising Hamiltonian and observed hospitalization. In statistical physics, y_i could be +1 and -1; in HOIST, $y_{i,t}$ can only be non-negative values. In this sense, the interpretations of the Ising Hamiltonian in statistical physics and HOIST are fundamentally different.

It's not very clear to me why the authors chose the Ising Hamiltonian to impose the spatial structure. What is the meaning of $\hat{y}_{i,t}^d$? It seems the authors use the Ising Hamiltonian to define another version of predicted hospitalization and want to fit it to observed hospitalization using the loss function L_d . Could the authors explain why hospitalization in nearby locations should follow this specific functional form? For instance, if the unit of $y_{i,t}$ is person, the unit of $\hat{y}_{i,t}^d$ is not person since there is an interaction term $y_{i,t} \times y_{j,t}$ on the right hand side. It would be great to justify that this functional form captures the qualitative relationship between hospitalizations across locations.

If the goal is to impose spatial constraints, there might be many possible functional forms. The Ising Hamiltonian is only one heuristic choice of such constraints.

2. The embedding of locations into a latent space is interesting. How is the distance in this latent space different from geographical distance? Are there any locations that are far away geographically but close in the latent space?

3. The generalization of the algorithm is challenging as many data sources are not publicly available, particularly healthcare information. Even the prediction target (hospitalization from medical claims,

which seems to be different from the CDC COVID-19 hospitalization data) is not publicly available. As a proof of concept for the new method, it is okay to use this dataset. But it might be difficult to deploy it in operational use. For example, will the data updated frequently in real time?

4. I am wondering how much training data are sufficient to train the HOIST algorithm. During the early phase of an outbreak, it may not have enough data to train the model.

5. One critical challenge in COVID-19 forecasting is the emergence of new variants that could lead to different disease severity. I am curious how the authors picked the training period and prediction period in the experiments. Could the algorithm use data from the Delta wave to predict hospitalization in the Omicron wave? Additionally experiments using different training and validation periods are needed to explore.

6. The authors should describe how to generate prediction uncertainty using HOIST. In Fig.7, it seems the predictive interval is very narrow.

7. The format of the manuscript does not follow the journal style (e.g., the abstract). There are too many figures in the manuscript and labels in many figures are illegible. The current format looks like a computer science conference paper.

Reviewer #2 (Remarks to the Author):

In this work, the authors propose a novel method to forecast COVID-19 hospitalizations at county scale in the US.

The model, HOIST, is a complex computational architecture based on several data streams that feed a deep learning model guided by an Ising-like energy potential.

The model is then used to forecast weekly hospitalizations in about 3,000 U.S counties between 2021-2022, after training it on data from 2020-2021.

Results show that the model outperforms several baselines and its results can be used to evaluate the cost-effectiveness of vaccinations in rural areas.

Overall, this is a solid study that brings some innovation in the realm of COVID-19 forecasting, a field that has been extensively investigated in the past 2 years.

I think the paper has its merits and it could be considered for publication in Nature Communications, if the authors could address the points I mention below.

Major strengths

- the study describes a novel method to forecast COVID-19 hospitalizations
- code and data are publicly available
- the method performs better than several benchmarks and sensitivity analysis has been adequately performed

Suggestions for improvement

- the paper introduces the Ising-like dynamics to take into account correlations among different spatial units that can be related to intrinsic demographic or socioeconomic features. The idea of modeling spatially connected subunits in epidemics is not new, though. Metapopulation modeling is a classic example, where subpopulations are typically connected by mobility flows. I would recommend the authors discuss similarities and differences between their approach and the most common metapopulation dynamics (see for instance: Colizza, V. and Vespignani, A., 2008. Journal of theoretical biology)

- as described in the paper, the model embeds the US counties into a multidimensional space through a large set of features. It would be interesting to see a representation of the US counties in this multidimensional space, to understand what regions get close to each other over time, even if they are geographically far. This could provide more insights into the hospitalization dynamics.

- maps of Figures 4, 5 and 8 are very hard to read for several reasons: 1) the colorbar and its labels are tiny, 2) counties are heterogeneous in size and populations so that large urban areas in the Northeast almost disappear, 3) the caption is not very informative since it is basically only a title.

I would recommend the authors revise these figures enlarging the maps and zooming into some large metro areas (Chicago, New York, Boston, Atlanta?) to provide additional spatial details.

Minor comments:

- references of the CDC MMWR are cited as webpages but they can be cited as papers, in the same way as other references

Reviewer #3 (Remarks to the Author):

This paper proposes a spatio-temporal model, namely, the HOIST, to predict the county-level number of hospitalizations during the COVID-19 pandemic in the US. The proposed model integrates an adaptive connection learning module, which captures the spatial similarities between locations, and an LSTM model, which captures the temporal hospitalization progression patterns. The classic Ising dynamics is used to model the inner-region factors and inter-region similarities simultaneously. The model was validated on 2000+ US counties. One contribution of this work is the use of similarities of geographical and socioeconomic properties between regions to improve the prediction performance. In general, the paper is well structured, but there is a lack of scientific merit to justify its publication in Nature Communications.

First off, hospitalization prediction has been extensively discussed in previous studies, not only COVID but also other infectious diseases. For example, reference 15 addresses the same problem with the same deep learning modeling approach. This study has some new incremental methodological contributions, but the scientific problem is not novel. This study presents a good application with some technological innovation, but it is difficult to say there is a significant scientific contribution here. It could be suitable for a CS conference or a technical journal.

Second, the Ising model is a classic statistical mechanics model. It is common to use it to model human dynamics. The author provided some justifications, which I agree with. But why it has to be Ising dynamics? Why not the other nonlinear dynamics or other machine learning modules to address this problem? So it is difficult to justify that this method is the best solution.

Third, according to Table 5, removing the adaptive connection learning module (HOIST-AC) does not significantly reduce the performance of the original HOIST model. HOIST-AC still significantly outperforms all baseline models and other ablation versions. This indicates that the similarity of geographical and socioeconomic properties between regions does not play an important role in hospitalization prediction. The new complication of the model brings little performance benefits – making this new complication and the main novelty of the paper less warranted.

Besides, many features are used in the adaptive connection learning module, but the authors did not comprehensively evaluate the specific contributions of each feature. For example, it remains unknown if the economic features, such as average annual income and unemployment rate, affect a region's hospitalization rate.

As for the medical claims and vaccination statistics, except for these widely accepted features (such as the number of patients older than 65 and high-risk conditions in CDC guidelines), many features (such as the brand of vaccines and the number of HIV-infected individuals) do not actually affect the prediction performance according to their analysis. I wonder if it is necessary to include these features in the prediction.

I also doubt the high performance of the proposed model. Limited information is provided about the settings of baseline models. For example, it is unclear how/if the static data is incorporated into baseline models. For these SIR/SEIR-based baseline models, no information is provided about the parameter settings in the underlying epidemic dynamics. It seems that the github link is only a visualization. There is no code or software to validate the results.

For the results in the section "Number needed to treat prediction with HOIST", because the number of regional hospitalizations is dynamic, it may not be reasonable to summarize the general patterns based on the information at a specific timestamp. Besides, according to the US CDC, the difference in death rates is much larger between age groups than races (<https://www.cdc.gov/coronavirus/2019-ncov/covid-data/investigations-discovery/hospitalization-death-by-age.html>). Why didn't the authors characterize the heterogeneity between regions by their age structures? Also, I did not find noticeable differences between NNT and adjusted NNT in Figure 8. The authors may want to highlight the significant difference between them.

Some minor issues:

- In equation (1), I did not find the definition of the K matrix.
- Results mentioned but not presented in the main text should be provided in the supplementary material for further reference, such as the E.F. weights of high-risk conditions.
- It is hard to read all figures with low-quality images and limited descriptions in captions.

We would like to thank the editors and reviewers for their careful review and constructive comments. We have revised our paper according to your suggestions. We also answered your questions below:

Reviewer 1

- 1. It's not very clear to me why the authors chose the Ising Hamiltonian to impose the spatial structure. What is the meaning of $\hat{y}_{i,t}^d$? It seems the authors use the Ising Hamiltonian to define another version of predicted hospitalization and want to fit it to observed hospitalization using the loss function L_d . Could the authors explain why hospitalization in nearby locations should follow this specific functional form? For instance, if the unit of $y_{i,t}$ is person, the unit of $\hat{y}_{i,t}^d$ is not person since there is an interaction term $y_{i,t} \times y_{j,t}$ on the right hand side. It would be great to justify that this functional form captures the qualitative relationship between hospitalizations across locations. If the goal is to impose spatial constraints, there might be many possible functional forms. The Ising Hamiltonian is only one heuristic choice of such constraints.**

The Ising Hamiltonian used in our work is a bit more general than the original definition, where y_i should be either +1 or -1. In our setting, we are predicting a continuous target. Since the prediction targets are normalized using the z-score normalization, these target values become positive or negative values. Moreover, we use the Ising dynamics as a regularization constraint instead of the only prediction equation in our model. The $\hat{y}_{i,t}^d$ can be rewrite as $\hat{y}_{i,t}^d = (e_i + \sum_j^N s_{ij} \hat{y}_{j,t}) \hat{y}_{i,t}$, which is essentially reweighting the original prediction $\hat{y}_{i,t}$ using inner-location factors and inter-location interactions. The underlying spatial assumption is that the future hospitalization cases of a location should be affected by the case count of similar locations (i.e., high s_{ij}).

We agree that the Ising Hamiltonian is not the only choice for real-world constraints. We chose the Ising dynamics because it can incorporate inter-region spatial relationships and inner-region factors and can handle these complex factors compared to naïve SIR and SEIR models. And it does show good experimental performances. We are inspired by previous sociology studies that use Ising models for human behavior tendencies¹. We hope our exploratory work can inspire future studies using other real-world dynamics. We have clarified this in the limitation section.

- 2. The embedding of locations into a latent space is interesting. How is the distance in this latent space different from geographical distance? Are there any locations that are far way geographically but close in the latent space?**

Thank you for your advice. We added a section for spatial embedding visualization in the supplementary material. We use the T-SNE to reduce the learned location embeddings $\mathbf{z}_{i,t}$ to two dimensions and plot the embeddings. The results show that HOIST can learn clustered spatial embeddings using the demographics data. HOIST can capture the similarity between large metropolis and small cities. We also select a few cities and find the nearest cities in the latent space. For New York County, the top 5 nearest counties are Queens NY, Kings NY, Wayne MI, King WA and Palm Beach FL. All five counties are close to New York or metropolis cities with more than 1 million populations.

Similarly, for Los Angeles County, the top 5 nearest counties are Kings NY, Queens NY, Philadelphia PA, Maricopa AZ, and New York NY. For small counties such as Schuyler IL, which only has a population of 7,544, the nearest locations are Washington IL, Greenwood KS and Sac IA, where Washington is next to Schuyler and the latter two locations both have populations less than 10,000. These results show that HOIST can not only extract geographical similarities between locations, but also can identify geographically distant but socio-economically similar locations. We also made an interactive T-SNE plot online at <https://v1xerunt.github.io/HOIST/>.

- 3. The generalization of the algorithm is challenging as many data sources are not publicly available, particularly healthcare information. Even the prediction target (hospitalization from medical claims, which seems to be different from the CDC COVID-19 hospitalization data) is not publicly available. As a proof of concept for the new method, it is okay to use this dataset. But it might be difficult to deploy it in operational use. For example, will the data updated frequently in real time?**

The claims database is updated weekly, so we plan to provide up-to-date predictions for future predictions. We use hospitalization counts from medical claims instead of CDC data because the CDC data sources seem to have more noise, such as missing data, which is difficult to train a stable model. Nevertheless, we also provide a version of our model built with only publicly available data sources, which can still achieve comparable performances and is still better than baseline models, as shown in Table 1 (HOIST-Claims). The model and data are available in the GitHub repo.

- 4. One critical challenge in COVID-19 forecasting is the emergence of new variants that could lead to different disease severity. I am curious how the authors picked the training period and prediction period in the experiments. Could the algorithm use data from the Delta wave to predict hospitalization in the Omicron wave? Additionally experiments using different training and validation periods are needed to explore.**

Thank you for your suggestion. To understand the temporal performance of our model, we already split the entire sequence by time into train, validation, and test sets with a 3:1:1 ratio. We also added a new temporal data split experiment to address your concern. We use a sliding window training setting by using 10 weeks of data for training and using the next 4 weeks of data for testing. We split the time from Sep 2020 to April 2022 into 7 periods and the models are tested in 7 testing time phases. The results show that HOIST consistently outperforms all baseline models in terms of MSE and MAE in all testing phases. We notice that some baseline models experienced low prediction performance on Feb 2021 and Jan 2022. This may be due to distribution shifts between training and testing data caused by emerging new variants of the COVID-19 virus. However, HOIST achieves lower prediction errors than baselines on these time phases.

- 5. I am wondering how much training data are sufficient to train the HOIST algorithm. During the early phase of an outbreak, it may not have enough data to train the model.**

From the data perspective, as we introduced in the last response, only 10 weeks of training data can achieve good prediction performance. This is useful at the emerging stage of a pandemic. From the model perspective, the number of parameters in HOIST is 110k, similar to naïve RNN models (LSTM 100k, GRU 90k) and less than most spatial-temporal prediction models (STAN 150k). So HOIST does not require additional data for training.

- 6. The authors should describe how to generate prediction uncertainty using HOIST. In Fig.7, it seems the predictive interval is very narrow.**

In Fig. 7, all models' standard deviations are generated by training models 5 times using different random initializations. As discussed in the limitation section, we observe that those baseline models sometimes have abnormally large prediction errors and standard deviations, but HOIST doesn't have this issue. Therefore, the predictive interval of HOIST is very narrow. This may be because the spatial connectivity learning module helps stabilize the predictions.

- 7. The format of the manuscript does not follow the journal style (e.g., the abstract). There are too many figures in the manuscript and labels in many figures are illegible. The current format looks like a computer science conference paper.**

Thank you for pointing it out. We have revised our paper to ensure it follows the journal's requirements. We have moved some nonessential content to the supplementary materials.

Reviewer 2

- 1. The paper introduces the Ising-like dynamics to take into account correlations among different spatial units that can be related to intrinsic demographic or socioeconomic features. The idea of modeling spatially connected subunits in epidemics is not new, though. Metapopulation modeling is a classic example, where mobility flows typically connect subpopulations. I would recommend the authors discuss similarities and differences between their approach and the most common metapopulation dynamics (see for instance: Colizza, V. and Vespignani, A., 2008. Journal of theoretical biology)**

We have added related metapopulation modeling works into the background section. Our work is different from existing metapopulation works in two aspects: (1). The underlying assumption of most metapopulation works is using population mobility flows to predict the pandemic. So they only utilize the spatial relationship between two adjacent locations. But the hospitalization prediction relies on not only geographic adjacency but also economic and healthcare resources similarity between locations. Our model can better handle both situations than these metapopulation analysis works. (2) Most metapopulation works still require a predefined location graph to model the spatial relationships. They largely ignore the underlying social, economic, and medical similarities, which may profoundly impact the interlocation relationships.

- 2. As described in the paper, the model embeds the U.S. counties into a multidimensional space through a large set of features. It would be interesting to see a representation of the U.S. counties in this multidimensional space, to understand what regions get close to each other over time, even if they are geographically far. This could provide more insights into the hospitalization dynamics.**

Thank you for your advice. We added a section for spatial embedding visualization in the supplementary material. We use the T-SNE to reduce the learned location embeddings $\mathbf{z}_{i,t}$ to two dimensions and plot the embeddings. The results show that HOIST can learn clustered spatial embeddings using the demographics data. HOIST can capture the similarity between large metropolis and small cities. We also select a few cities and find the nearest cities in the latent

space. For New York County, the top 5 nearest counties are Queens NY, Kings NY, Wayne MI, King WA and Palm Beach FL. All five counties are close to New York or metropolis with more than 1 million populations. Similarly, for Los Angeles County, the top 5 nearest counties are Kings NY, Queens NY, Philadelphia PA, Maricopa AZ and New York NY. For small counties such as Schuyler IL, which only has a population of 7,544, the nearest locations are Washington IL, Greenwood KS and Sac IA, where Washington is next to Schuyler and the latter two locations both have populations less than 10,000. These results show that HOIST can not only extract geographical similarities between locations, but also can identify geographically distant but socio-economically similar locations. But it is worth noting that the learned spatial similarities won't change with time since we use static data to model them. This can be further improved by using more data resources, such as dynamic population mobility data in future works. We also made an interactive T-SNE plot online at <https://v1xerunt.github.io/HOIST/>.

- 3. Maps of Figures 4, 5 and 8 are very hard to read for several reasons: 1) the colorbar and its labels are tiny, 2) counties are heterogeneous in size and populations so that large urban areas in the Northeast almost disappear, 3) the caption is not very informative since it is basically only a title. I would recommend the authors revise these figures enlarging the maps and zooming into some large metro areas (Chicago, New York, Boston, Atlanta?) to provide additional spatial details.**

We have enlarged these figures and increased the label font sizes. We also have added more descriptions in the captions. We also encourage readers to check out our interactive versions of those figures at <https://v1xerunt.github.io/HOIST/>.

- 4. References of the CDC MMWR are cited as webpages but they can be cited as papers, in the same way as other references**

We added these webpages to references.

Reviewer 3

- 1. First off, hospitalization prediction has been extensively discussed in previous studies, not only COVID but also other infectious diseases. For example, reference 15 addresses the same problem with the same deep learning modeling approach. This study has some new incremental methodological contributions, but the scientific problem is not novel. This study presents a good application with some technological innovation, but it is difficult to say there is a significant scientific contribution here. It could be suitable for a C.S. conference or a technical journal.**

Our contribution is introducing new real-world dynamics into deep learning models, which have been proven to help guide the model learning process. Our work is the first model to use real-world dynamics to predict hospitalization cases. Compared with other existing works, the proposed model can extract and utilize both inter-region similarities and complex inner-region influence factors. Our model achieves significantly higher prediction performance than naive RNNs (in reference 15) and other state-of-the-art epidemiology prediction models. Our analysis

of the NNT, cost ratio, and E.F. weights also provide valuable clinical insights for policymakers and clinicians.

- 2. Second, the Ising model is a classic statistical mechanics model. It is common to use it to model human dynamics. The author provided some justifications, which I agree with. But why it has to be Ising dynamics? Why not the other nonlinear dynamics or other machine learning modules to address this problem? So it is difficult to justify that this method is the best solution.**

We agree that the Ising dynamics is not the only choice for real-world constraints. We choose the Ising dynamics because it can incorporate inter-region spatial relationships and inner-region factors and handle these complex factors compared to naïve SIR and SEIR models. And it does show good experimental performances. We are inspired by previous sociology studies that use Ising models for human behavior tendencies¹. We hope our exploratory work can inspire future studies using other real-world dynamics. We have clarified this in the limitation section.

- 3. Third, according to Table 5, removing the adaptative connection learning module (HOIST-AC) does not significantly reduce the performance of the original HOIST model. HOIST-AC still significantly outperforms all baseline models and other ablation versions. This indicates that the similarity of geographical and socioeconomic properties between regions does not play an important role in hospitalization prediction. The new complication of the model brings little performance benefits – making this new complication and the main novelty of the paper less warranted.**

The adaptative connection module is a part of the Ising dynamics, our major contribution. Learning inter-region relationship and inner-region influence factors together forms the final Ising dynamic loss. Indeed, the A.C. module has less performance impact than the E.F. weight learning module, but HOIST-AC achieves 91% higher MSE than HOIST. Besides, as discussed in the limitation section, we found that HOIST does not suffer from the unstable training process like many other baselines. This may be because learning spatial similarities between locations help stabilize the learning process and improve numerical stability.

To further prove the effectiveness of the adaptative connection learning module, we added a section for spatial embedding visualization in the supplementary material. We use the T-SNE to reduce the learned location embeddings $\mathbf{z}_{i,t}$ to two dimensions and plot the embeddings. The results show that the A.C. module can help learn clustered spatial embeddings using the demographics data. These results show that HOIST can not only extract geographical similarities between locations, but also identify geographically distant but socio-economically similar locations.

- 4. Besides, many features are used in the adaptative connection learning module, but the authors did not comprehensively evaluate the specific contributions of each feature. For example, it remains unknown if the economic features, such as average annual income and unemployment rate, affect a region's hospitalization rate.**

The feature contributions can be evaluated using other techniques, such as the Shapley value. However, the demographic and socioeconomic feature contributions to spatial connectivity are not the focus of this paper, as they are static data and do not change with time. We aim to learn a comprehensive distance metric between locations to evaluate geometric, social, and economic similarities. Incorporating more dynamic data sources to model spatial connectivity more

comprehensively and interpretability analysis could be our future work. We have clarified this in the limitation section.

- 5. As for the medical claims and vaccination statistics, except for these widely accepted features (such as the number of patients older than 65 and high-risk conditions in CDC guidelines), many features (such as the brand of vaccines and the number of HIV-infected individuals) do not actually affect the prediction performance according to their analysis. I wonder if it is necessary to include these features in the prediction.**

According to our results, removing either disease condition features (HOIST-Risk) or vaccination statistics (HOIST-Vaccination) will lead to higher prediction errors. The high-risk condition features are selected from the Charlson comorbidity index, widely used in the outcome analysis research for COVID-19 patients^{2,3}. These features are proven to have significant impacts on patients' outcomes. Besides, using these features also enable us to conduct further analysis, such as the effect of different vaccination brands and using vaccination statistics to compute NNTs. We believe including these features can reduce prediction errors and help discover new public health insights. We have clarified this in our paper.

- 6. I also doubt the high performance of the proposed model. Limited information is provided about the settings of baseline models. For example, it is unclear how/if the static data is incorporated into baseline models. For these SIR/SEIR-based baseline models, no information about the parameter settings in the underlying epidemic dynamics is provided. It seems that the github link is only a visualization. There is no code or software to validate the results.**

For the spatial-temporal prediction models (i.e., ColaGNN, ACTS, CovidGNN, STAN), we use the static data to build the location graph or calculate the location similarities in their algorithms. For the GRU and LSTM models, we concatenate the static data with the original inputs at each timestep to the model. The SEIR model cannot take the static location data as inputs. For the DELPHI-SEIR model, we use the deployed version of the DELPHI-SEIR model (i.e., DELPHI-SEIR V4) on their website (<https://www.covidanalytics.io/>). They provided a recommended optimal parameter list in their code repo and we finetuned some of the key parameters on the validation set (e.g., IncubeD, RecoverID, RecoverHD). We have clarified these in our paper.

We have provided the complete model code, data, training logs, and testing results in the publicly available github repo (<https://github.com/v1xerunt/HOIST>, this link is different from the visualization website link) as well as the zip file submitted to the journal. We have also released two versions: one that uses claims data and another that only requires publicly available data. The model built with only publicly available data sources can also achieve comparable performances and is still better than baseline models, as shown in Table 1 (HOIST-Claims). We have emphasized this by adding a 'Materials & Correspondence' section.

- 7. For the results in the section "Number needed to treat prediction with HOIST", because the number of regional hospitalizations is dynamic, it may not be reasonable to summarize the general patterns based on the information at a specific timestamp. Besides, according to the US CDC, the difference in death rates is much larger between age groups than races (<https://www.cdc.gov/coronavirus/2019-ncov/covid-data/investigations-discovery/hospitalization-death-by-age.html>). Why didn't the authors characterize the heterogeneity between regions by their age structures? Also, I did not find noticeable**

differences between NNT and adjusted NNT in Figure 8. The authors may want to highlight the significant difference between them.

Thank you for your suggestion. The race-adjusted NNT and hospitalization NNT do not have significant differences but we find that locations with high cost ratios also have high race-adjust ratios. For example, in Butte, Idaho, the cost ratio is 60.3 and the race-adjust ratio is 1.78. This may indicate significant healthcare disparities for different races in those locations. More lives could be saved by improving vaccination in these locations. We provided more detailed discussions in the paper.

Exploring age-adjusted NNT is a good idea in general. However, due to the lack of access to the granular age information, we are not able to calculate this age-adjusted NNT precisely. Since the age distribution of hospitalized patients is probably very similar across counties (e.g., hospitalizations are concentrated in older population), age-adjust NNT are unlikely to change relative orders compared to the original NNT measures. Nevertheless, we hope we can improve this in future by incorporating more granular age information of hospitalized patients. We have added this in the limitation section.

8. In equation (1), I did not find the definition of the K matrix.

The K matrix is the population demographics M and the economics and healthcare statistics M (i.e., $\forall K \in \{M, E\}$ under the sum symbol). The idea is to calculate the latent distance based on all socioeconomic features. We have made further clarifications on this in our paper.

9. Results mentioned but not presented in the main text should be provided in the supplementary material for further reference, such as the E.F. weights of high-risk conditions.

The E.F. weights of high-risk conditions are described in the ‘Analysis of learned external fields’ section. We find the feature with the highest E.F. weight is the number of patients older than 65. The top 5 conditions are: renal disease, dementia, immunodeficiency, malignancy, and chronic lung disease. The results are consistent with CDC guidelines, and we do not observe statistically significant differences in these five features.

10. It is hard to read all figures with low-quality images and limited descriptions in captions.

We have enlarged these figures and increased the label font sizes. We also have added more descriptions in the captions.

References

1. Godoy-Lorite, A. & Jones, N.S. Inference and influence of network structure using snapshot social behavior without network data. *Science Advances* **7**, eabb8762 (2021).
2. Kuswardhani, R.T., *et al.* Charlson comorbidity index and a composite of poor outcomes in COVID-19 patients: A systematic review and meta-analysis. *Diabetes & Metabolic Syndrome: Clinical Research & Reviews* **14**, 2103-2109 (2020).
3. Christensen, D.M., *et al.* Charlson comorbidity index score and risk of severe outcome and death in Danish COVID-19 patients. *Journal of general internal medicine* **35**, 2801-2803 (2020).

REVIEWER COMMENTS

Reviewer #1 (Remarks to the Author):

The authors have addressed most of my questions. I appreciate the efforts. However, I would like to emphasize again on the issue of prediction uncertainty. For operational forecasting, over-confident predictions with narrow predictive intervals can be a problem (precise but not accurate). The evaluation metrics used in the study (MSE, MAE, R2, and CCC) cannot reflect the performance in terms of uncertainty. I would recommend the authors to check out some recent papers on collaborative real-time infectious disease forecasting and see how predictions were evaluated in the community of public health (e.g., Cramer et al. PNAS 119 (15), e2113561119 (2022)). For instance, the coverage of prediction interval can quantify how many targets were captured by a certain prediction interval. Ideally, the X% prediction interval should cover X% targets. The HOIST model is probably too heavy to run for a large number of realizations to get the precise prediction interval. But the authors can still get some uncertainty quantification (e.g., STD) and convert those to prediction intervals under some assumed distributions (e.g., Gaussian). I think there should be at least one metric to quantify the uncertainty (such as the prediction interval coverage) and it is okay the coverage is not ideal, like the case of real-time COVID-19 forecasting.

Reviewer #2 (Remarks to the Author):

I thank the authors for addressing my remarks in their revised version of the manuscript. The paper has significantly improved and I believe it can be accepted for publication.

Reviewer #3 (Remarks to the Author):

The authors have addressed my technical comments. Many thanks to the authors' efforts.

I hesitate to fully agree with the scientific merit of the study to justify publication in Nat Comm, but I would not firmly object to the publication of the paper either.

We would like to thank the editors and reviewers for their thorough review and insightful feedback. The paper has been revised based on their suggestions. The following addresses their questions:

Reviewer 1

- 1. The authors have addressed most of my questions. I appreciate the efforts. However, I would like to emphasize again on the issue of prediction uncertainty. For operational forecasting, over-confident predictions with narrow predictive intervals can be a problem (precise but not accurate). The evaluation metrics used in the study (MSE, MAE, R2, and CCC) cannot reflect the performance in terms of uncertainty. I would recommend the authors to check out some recent papers on collaborative real-time infectious disease forecasting and see how predictions were evaluated in the community of public health (e.g., Cramer et al. PNAS 119 (15), e2113561119 (2022)). For instance, the coverage of prediction interval can quantify how many targets were captured by a certain prediction interval. Ideally, the X% prediction interval should cover X% targets. The HOIST model is probably too heavy to run for a large number of realizations to get the precise prediction interval. But the authors can still get some uncertainty quantification (e.g., STD) and convert those to prediction intervals under some assumed distributions (e.g., Gaussian). I think there should be at least one metric to quantify the uncertainty (such as the prediction interval coverage) and it is okay the coverage is not ideal, like the case of real-time COVID-19 forecasting.**

We have added a section on prediction uncertainty analysis to the supplementary material and included 90% confidence intervals in all county-level and state-level predictions. These intervals are calculated using conformal methods, which are robust and require no assumptions about the underlying data distribution or model. Results show that the HOIST model achieved 95% coverage rate on the validation set and 89.3% coverage rate on the test set. Compared to other baselines, HOIST showed a narrower prediction interval, indicating more precise and accurate predictions with higher confidence. This information can aid in making informed and timely decisions in real-world situations.

References

1. Angelopoulos, A.N. & Bates, S. A gentle introduction to conformal prediction and distribution-free uncertainty quantification. *arXiv preprint arXiv:2107.07511* (2021).

REVIEWERS' COMMENTS

Reviewer #1 (Remarks to the Author):

The authors have addressed my question.